# Pedestrians’ Perceptions of Motorized Traffic Variables in Relation to Appraisals of Urban Route Environments

**DOI:** 10.3390/ijerph20043743

**Published:** 2023-02-20

**Authors:** Dan Andersson, Lina Wahlgren, Karin Sofia Elisabeth Olsson, Peter Schantz

**Affiliations:** 1The Research Unit for Movement, Health and Environment, Department of Physical Activity and Health, The Swedish School of Sport and Health Sciences (GIH), 114 86 Stockholm, Sweden; 2Department of Public Health and Clinical Medicine, Section of Sustainable Health, Umeå University, 901 87 Umeå, Sweden

**Keywords:** walking, active transportation, motorized vehicle speed, flow, exhaust fumes, noise, unsafe–safe traffic, hinders–stimulates walking, environmental unwellbeing–wellbeing

## Abstract

It is important to examine how motorized traffic variables affect pedestrians along a gradient from rural to inner urban settings. Relations between pedestrians’ perceptions of four traffic variables and appraisals of route environments as hindering–stimulating for walking as well as unsafe–safe for reasons of traffic, were therefore studied in the inner urban area of Stockholm, Sweden (*n* = 294). The pedestrians rated their perceptions and appraisals with the Active Commuting Route Environment Scale (ACRES). Correlation, multiple regression, and mediation analyses were used to study the relationships between the traffic variables and the outcome variables. Noise related negatively to both hindering–stimulating for walking, and to unsafety–safety for traffic reasons. Vehicle speed related negatively to unsafety–safety for traffic reasons. Furthermore, vehicle speed protruded as an important origin of the deterring effects of traffic among those who commute by foot. The study shows the value of both partial and simultaneous analyses of the effect of all four traffic variables in relation to outcome variables relevant for walking.

## 1. Introduction

Walking has been a major source of physical activity in the history of mankind. Today, its benefits on health are well recognized [1,2,3]. At the same time, a considerable portion of the adult population is physically inactive [4], which calls for a search for pathways to promote physical activity. From that perspective, walking is of particular interest since it can be easily undertaken and demands no specific equipment. It is, however, important that the external environments support this form of locomotion, both as a behaviour and through creating environmental wellbeing.

Due to the introduction of motorized vehicles on a large scale, from the second world war and onwards, there has been a dramatic change in the route environments amongst those who walk. It is very important to examine how these altered traffic landscapes affect pedestrians along a gradient from rural to inner urban settings.

Many environmental assessment tools related to walking have been developed. Examples of these include the Neighborhood Environment Walkability Scale (NEWS) [5], the Twin Cities Walking Survey (TCWS) [6], and the Saint Louis Environment and Physical Activity Instrument [7]. They all have questions about aspects of the built and traffic environments, primarily with four-point response scales of the Likert type.

However, to understand the effect of individual traffic variables on walking behaviour and/or wellbeing while walking is challenging. Multiple variables, that capture different aspects of the environment, e.g., vehicle speed, vehicle flow, noise, and exhaust fumes, should preferably be included in the analyses. This requirement has, in many cases, not been fulfilled, e.g., [8,9,10]. Hence, diverse, and sometimes contradictory findings have been reported regarding transportation walking. For example, one study reported that participants preferred streets with little traffic [11], while in another study, traffic volume did not show an association with walking [12]. It is also evident from that research field that one might walk regardless of an unsupportive external environment, e.g., for economic reasons, or that areas are not serviced by public transport. Thus, walking behaviours may not be related to traffic environments.

Furthermore, the lack of a common nomenclature makes comparisons between studies complicated. For example, Humpel et al., [8] used the following items: “*Crossing busy roads is a big problem*” and “*Traffic makes it dangerous or unpleasant*”. How should these items be interpreted in relation to the traffic variables?

Given that, a more fruitful way could be to analyse how different traffic variables are perceived, and route environments appraised, by those who regularly walk in those settings. For that purpose, we suggest a conceptual framework involving three interconnected levels (Figure 1). In order to explore such relationships between active transportation and the route environment, the Active Commuting Route Environment Scale (ACRES) has been developed and evaluated [13,14]. The ACRES is a self-report tool which assesses cycling and pedestrian commuters´ perceptions and appraisals of their individual commuting route environments.

This conceptual framework is based on several inputs. One is long-term educational experience by one of the authors (PS). Further support comes from the scientific literature. Safety (for different reasons) has been reported as an important issue among those who walk [11,16,17]. In addition, a need for the supplementary outcome of hinders–stimulates walking has been identified [18]. Furthermore, Panter et al. evaluated associations between shifts in perceptions of the route environment, and changes in active commuting [19]. When it became less pleasant to walk, there was a drop in the amount of walking. This refers to the walking behavior in the final level of the model. Physical activity can affect wellbeing [20], which refers to the psychological effects in the model. It is important to separate that effect on wellbeing from how walking routes can affect environmental wellbeing. An example of the latter is that traffic volume and speed has been described as both as a barrier for walking and affecting wellbeing negatively [9].

As stated by Anciaes et al., [9] many researchers have only used one variable as a proxy for traffic. That can be a reasonable strategy, but only if it is based on analyses supporting the choice of such proxies. In our mind, this prompt attempts to sort out the relationships between four traffic variables: *speeds of motor vehicles*, *flow of motor vehicles*, *noise*, and *exhaust fumes*, and how they relate to the outcomes *hinders*–*stimulates walking* and *unsafe*–*safe traffic* (Figure 2).

We consider *speeds of motor vehicles* and *flow of motor vehicles* as basic traffic variables and *noise* and *exhaust fumes* as intermediate outcomes. This is also reflected by the fact that models for calculating noise depends on both flow and speeds of motor vehicles [21,22].

In this explorative and relational study, our focus is therefore on the relationships between: (a) the four traffic variables, (b) the basic variables and the intermediate outcomes, respectively, as well as (c) how the four traffic variables, and combinations of them, relate to the outcome variables. To our knowledge there exist no previous studies of these issues. We have studied male and female walking commuters (*n* = 294) rating their commuting routes using ACRES in the inner urban area of Stockholm, Sweden.

The rationales for these foci are to understand these relations as such, and to create a basis for how study protocols, audit tools and questionnaires can be outlined, possibly through choosing relevant proxy variables. Furthermore, the present study can assist in understanding results of studies which have used less than four traffic variables in their ambition to highlight how traffic affects pedestrians. Research such as this can also provide urban and traffic planners, and others, with important information on how to create attractive route environments, hence, contributing to active transportation and improved public health.

## 2. Method

### 2.1. Procedure and Participants

This study is part of the Physically Active Commuting in Greater Stockholm (PACS) research project. Individuals who walk or cycle to their place of work or study, were recruited via two major newspapers in Stockholm, in May and June 2004. The participants had to be at least 20 years old, reside in the County of Stockholm (excluding Norrtälje Municipality) (see Figure 3), and actively commute the entire distance at least once a year.

The advertisement resulted in 2148 individuals volunteering to participate. A first survey, the Physically Active Commuting in Greater Stockholm Questionnaire (PACS Q1) was distributed in September 2004 (response frequency = 94%). A second survey (PACS Q2) was distributed in May 2005 (response rate = 92%). The respondents commuted in the inner urban or suburban–rural areas of Greater Stockholm, or in both of these settings (see Figure 3 and Figure 4). For additional information about the recruitment process, see Wahlgren and Schantz [13,23].

Advertisement recruitment, as a sampling method, has been compared to street recruitment on cycle commuters regarding ratings of route environments [13]. It was hypothesized that the street-recruitment strategy could represent those who actively commuted with greater certainty than the advertisement-recruited sample. Overall, the ratings indicated a good correspondence between the two groups. Furthermore, the characteristics of the two samples, with respect to background factors such as age, employment, and education, showed reasonable compliance [24].

In this study, we have solely utilised data from pedestrian commuting in the inner urban area. Some of them (26%) commuted in the suburban area as well, but only data from the inner urban area has been used. A previous study by Wahlgren and Schantz [13], has indicated that cycle commuting in both areas, does not affect ratings in any of the areas.

A total of 294 participants (77% women) were included in the analyses: 56.5% were pedestrians and 43.5% dual-mode commuters, i.e., individuals who alternate between walking and cycling. For descriptive characteristics of the participants, see Table 1, and for characteristics of their walking behavior, see Table A1, Table A2 and Table A3 in Appendix A. The Ethics Committee North of the Karolinska Institute at the Karolinska Hospital approved the study (Dnr 03-637). The participants gave their informed consent.

### 2.2. Descriptive Characteristics of the Participants

Descriptive characteristics of the participants were obtained from PACS Q1 and Q2, see Table 1.

### 2.3. The Physically Active Commuting in Greater Stockholm Questionnaires (PACS Q1 and Q2)

PACS Q1 and PACS Q2 are self-report surveys in Swedish. The questionnaires include background questions and questions related to active commuting. Both surveys, including English versions, can be found as supporting information in Schantz et al. [25]. In total, they consist of 103 items.

#### The Active Commuting Route Environment Scale (ACRES)

To investigate the relationships between perceptions and appraisals of active commuters and the route environment, the Active Commuting Route Environment Scale (ACRES) was created [13,14]. The ACRES is included in PACS Q2. The pedestrian version consist of 13 items and each item considers both the inner urban area, and the suburban—rural areas. All items have two parallel response lines, one for each area. The scales have adjectival opposites (see Table 2) and a neutral mid-position.

The participants were encouraged to rate their experience of their route environments based on their commuting during the preceding two weeks. A more in-depth portrayal of the development of the scale, as well as its validity and reliability for cyclists, has been reported by Wahlgren et al. [14] and by Wahlgren and Schantz [13].

### 2.4. Study Area

The commuting environments are located in the inner urban area of Stockholm, Sweden. The city itself stretches across several islands where Lake Mälaren flows into the Baltic Sea. When data was collected, the population was approximately 1.9 million. The inner urban area, in our geographical division of the city, includes the following areas: Gamla stan, Södermalm, Kungsholmen, Vasastan, Norrmalm and Östermalm (Figure 4). Blocks are organized in grid-like streetscapes, constituting a different environment compared to the suburban and rural parts of Stockholm County. Throughout the city there are green and blue spaces, squares, and historic buildings.

Within these inner urban areas, several different roads exist, e.g., local streets, collector roads, minor and major arterial roads. Typically, the arterials have bidirectional traffic, whereas local streets normally have unidirectional traffic. Sidewalks are always present on both sides of the streets. The motor vehicles were cars, lorries and buses with internal combustion engines. Data was collected prior to the introduction of electric motor vehicles, electrically-assisted bikes, and e-scooters. Furthermore, pedestrians did not use headphones/earphones nearly as much as they do today.

In a sample of local streets, a collector road, and minor arterial roads, the vehicle flow on weekdays varied between 222–1726 vehicles per hour in both directions during the morning rush hour (7–9 a.m.), and between 291–1827 vehicles per hour in both directions during the evening rush hour (4–6 p.m.). The corresponding vehicle speeds ranged between 24.4–38.4 and 24.8–40.9 km per hour in the morning and evening rush hours, respectively. For a more in-depth portrayal of vehicle flows and vehicle speeds on different roads, at different times, in these inner urban areas, as well as where the measurements took place, see Figure A1, Figure A2 and Figure A3 in Appendix A. When considering that the pedestrians also walked along local streets with less traffic, it can be concluded that a variety of traffic settings formed the basis for the ratings. The great majority of the respondents walked in good natural light conditions (Table A2 and Table A3) cf. [26]. 

### 2.5. Statistical Analyses

Data were entered into the Statistical Package for the Social Sciences (SPSS) and analysed in version 27.0 (IBM SPSS Inc., Somers, NY, USA). Data from PACS Q2, which include the ACRES, were checked for accuracy. Some respondents were excluded, predominantly because of incomplete or incorrect ACRES data. Data from those with no missing ACRES values were used for the route environmental analyses.

Differences between ratings of ACRES with respect to gender were examined with independent *t*-tests. Correlation analyses between the predictor and outcome variables were assessed with Pearson’s correlation coefficient (r). The correlations were, in absolute values, r ≤ 0.774. Correlations between the background variable *age* and the predictor variables of motor traffic were, in absolute values, r ≤ −0.137. According to Tabachnik and Fidell, [27] multicollinearity can be a problem with correlations ≥ 0.90 whereas Field, [28] suggests that attention should be paid to predictors with a correlation above 0.80.

Multiple regression analysis was used to examine the relationships between the predictors and the outcomes. 

The values from the multiple regression analyses are displayed as y-intercepts, unstandardized coefficients B, and their 95% confidence intervals, as well as adjusted R square for the overall models. A statistical level corresponding to at least *p* < 0.05 was chosen to indicate significance. Prior to running the analyses, the linearity between the variables was assessed visually with the help of boxplots, error bars and scatterplots. The variables showed acceptable linearity.

Mediation analyses using PROCESS, a macro to SPSS, were run in models where a direct or indirect effect was hypothesized to occur. PROCESS, written by Professor Andrew F. Hayes, can be downloaded from www.processmacro.org [29] (accessed on 1 February 2022). To test the indirect effect, we used 5000 bootstrap samples to generate a confidence interval.

Four background variables were included in the multiple regression analyses and in the mediation analyses: *sex* (binary variable), *age* (continuous variable), *education* (binary variable: educated at university level or not educated at university level) and *income* (categorical variable coded into three categories: ≤25,000 SEK/month, 25,001–30,000 SEK/month and ≥30,001 SEK/month; Swedish crowns (SEK) 2005: €1 ≈ 9 SEK; US$1 ≈ 8 SEK). Only significant background variables are reported in the multiple regression ana-lyses.

The variance inflation factor (VIF) was used to assess multicollinearity. All models with predictor and outcome variables, including the four background variables, were checked (all values ≤ 3.30, mean: 1.22) indicating no dilemma [28]. Cook’s distance was used to identify extreme data cases. No such cases were found in either of the models, (all values ≤ 0.086, mean: 0.004) indicating no problem [28]. The top limit for inclusion of standardized residuals was set to ± 3 SD. A total of 30 individual cases (in 22 models) had a standardized residual above the top limit. Since they were few, relatively close to the limit for inclusion, and had no problems with extreme data cases, they were included in the multiple regression analyses. A table with the VIF, the Cook’s distance and the standardized residuals can be found in Appendix A (Table A4).

## 3. Results

### 3.1. Perceptions of the Environmental Variables in Males and Females

Levels of the outcome variables *hinders*–*stimulates walking* and *unsafe*–*safe traffic* as well as of the predictor variables *vehicle speed, vehicle flow*, *noise* and *exhaust fumes* are given in Table 3. A significant difference between males and females was noted in *vehicle speed*.

### 3.2. Correlations between the Environmental Variables

There were positive correlations between all predictor variables (range: 0.462–0.774), whereas they correlated negatively with the outcome variables *hinders*–*stimulates walking* (range: −0.210–−0.328) and *unsafe*–*safe traffic* (range: −0.183–−0.238) (Table 4, Figure 5 and Figure A4 in Appendix A).

**Table 4 ijerph-20-03743-t004:** Correlation matrix for the environmental variables (r).

	Hinders–Stimulates Walking	Unsafe–Safe Traffic	Vehicle Speed	Vehicle Flow	Noise	Exhaust Fumes
Hinders– stimulates walking	—					
Unsafe–safe traffic	0.313 *	—				
Vehicle speed	−0.210 *	−0.222 *	—			
Vehicle flow	−0.284 *	−0.183 *	0.612 *	—		
Noise	−0.328 *	−0.238 *	0.549 *	0.774 *	—	
Exhaust fumes	−0.274 *	−0.189 *	0.462 *	0.728 *	0.737 *	—

Note: * = Pearson´s correlation coefficient (r) is significant at the 0.01 level.

### 3.3. Relations between the Predictor Variables

The relations between the predictor variables were analysed with simultaneous multiple regression analyses. All regression coefficients were positive (range: 0.508–0.770; *p* < 0.001) (Table 5 and Figure 5).

### 3.4. Relations between the Basic Variables Vehicle Speed and Vehicle Flow and the Intermediate Outcomes Noise and Exhaust Fumes

Both vehicle speed and vehicle flow were significant in relation to the intermediate outcome noise. Only vehicle flow was significant in relation to exhaust fumes. For all values, see Table 6 and Figure 6.

**Table 6 ijerph-20-03743-t006:** Relations between the basic variables *vehicle speed* and *vehicle flow* and the intermediate outcomes *noise* and *exhaust fumes*.

Intermediate Outcome	y-Intercept (95% CI)	*p*-Value	Predictor	Regression Coefficient, Unstandardized B (95% CI)	*p*-Value	Adj. R^2^
Noise	1.40 (−0.19–2.99)	0.083	Vehicle speed	0.128	0.011	0.602
				(0.030–0.226)		
			Vehicle flow	0.632	<0.000	
				(0.549–0.715)		
Exhaust fumes	3.60 (1.77–5.43)	<0.000	Vehicle speed	0.026	0.657	0.525
				(−0.088–0.139)		
			Vehicle flow	0.668	<0.000	
				(0.573–0.764)		

Notes: The background variables *sex*, *age*, *education*, and *income* were included in the analysis. There were no significant background variables.

### 3.5. Relations between the Predictor Variables and the Outcome Hinders–Stimulates Walking 

All regression coefficients were significant and negative; range: −0.198–−0.296, with the lowest value for *vehicle speed* and the highest for *noise*. For all values see Table A5 and Figure A4 in Appendix A.

### 3.6. Relations between Combinations of Predictor Variables and the Outcome Hinders–Stimulates Walking

In the first combination, *vehicle flow* was negatively related with *hinders–stimulates walking*, and in the second combination *noise* had the corresponding role. When all traffic variables were included as predictors, only *noise* was negatively related to *hinders–stimulates walking* (Table 7, Figure 7).

**Table 7 ijerph-20-03743-t007:** Relations between combinations of predictor variables and the outcome *hinders–stimulates walking*.

Outcome	y-Intercept (95% CI)	*p*-Value	Predictor	Regression Coefficient, Unstandardized B (95% CI)	*p*-Value	Adj. R^2^
Hinders–stimulates walking	11.4 (9.20–13.6)	<0.000	Vehicle speed	−0.058	0.398	0.081
				(−0.194–0.077)		
			Vehicle flow	−0.194	0.001	
				(−0.308–−0.080)		
Hinders–stimulates walking	11.6 (9.54–13.7)	<0.000	Noise	−0.268	<0.000	0.111
				(−0.413–−0.122)		
			Exhaust fumes	−0.037	0.600	
				(−0.177–0.102)		
Hinders–stimulates walking	11.8 (9.61–14.0)	<0.000	Vehicle speed	−0.026	0.700	0.106
				(−0.162–0.109)		
			Vehicle flow	−0.024	0.771	
				(−0.184–0.137)		
			Noise	−0.242	0.006	
				(−0.415–−0.070)		
			Exhaust fumes	−0.026	0.732	
				(−0.176–0.124)		

Notes: The background variables *sex*, *age*, *education*, and *income* were included in the analysis. There were no significant background variables.

### 3.7. Relations between the Predictor Variables and the Outcome Unsafe–Safe Traffic 

All relations between each single traffic variable and the outcome *unsafe–safe traffic* were negative; range: −0.170–−0.247, with the lowest value for *vehicle flow* and the highest for *noise* (Table A6 and Figure A4 in Appendix A).

### 3.8. Relations between Combinations of Predictor Variables and the Outcome Unsafe–Safe Traffic 

In the first combination, *vehicle speed* was negatively related with *unsafe–safe traffic*, and in the second combination, *noise* had the equivalent role. When all traffic variables were included as predictors, both *vehicle speed* and *noise* were significantly related to the outcome (Table 8, Figure 8 and Figure 9).

**Table 8 ijerph-20-03743-t008:** Relations between combinations of predictor variables and the outcome *unsafe–safe traffic*.

Outcome	y-Intercept (95% CI)	*p*-Value	Predictor	Regression Coefficient, Unstandardized B (95% CI)	*p*-Value	Adj. R^2^
Unsafe–safe traffic	12.8 (10.2–15.4)	<0.000	Vehicle speed	−0.190	0.019	0.040
				(−0.349–−0.032)		
			Vehicle flow	−0.073	0.285	
				(−0.206–0.061)		
Unsafe–safe traffic	12.5 (10.0–15.0)	<0.000	Noise	−0.229	0.010	0.045
				(−0.402–−0.056)		
			Exhaust fumes	−0.024	0.779	
				(−0.189–0.142)		
Unsafe–safe traffic	13.2 (10.6–15.8)	<0.000	Vehicle speed	−0.163	0.045	0.052
				(−0.322–−0.004)		
			Vehicle flow	0.083	0.388	
				(−0.106–0.272)		
			Noise	−0.208	0.046	
				(−0.411–−0.004)		
			Exhaust fumes	−0.037	0.684	
				(−0.213–0.140)		

Notes: The background variables *sex*, *age*, *education*, and *income* were included in the analysis. There were no significant background variables.

### 3.9. Mediation

Mediation analyses were run in models where a possible mediating effect was of interest to investigate (Table 9). In addition to previously analysed variables, a composite variable was created by multiplying *vehicle flow* with *vehicle speed* since this combination possibly could affect the outcome variables. Only mediated effects (*p* < 0.05) corresponding to 40% or more are commented upon. An indirect effect of *vehicle speed* on *noise* is mediated by *vehicle flow* (78%), and an indirect effect of *vehicle speed* on *exhaust fumes* is mediated by *vehicle flow* (95%). *Noise* mediates the effect of *vehicle flow* and *vehicle speed* to *hinders–stimulates walking* (the indirect effects were 81% and 82% respectively). *Noise* mediates the effect of *vehicle flow* and *vehicle speed* to *unsafe–safe traffic* (the indirect effects were 104% and 42% respectively).

### 3.10. A Graphic Illustration of Important Pathways Based on the Commuting Pedestrians’ Perceptions and Appraisals of Their Route Environments

When adding mediated effects (*p* < 0.05) corresponding to 40% or more to Figure 9, the following aggregated illustration emerges, see Figure 10.

## 4. Discussion

One of the main results of this study, on four motorized traffic variables, was that *noise* was negatively related to both of the outcomes *hinders–stimulates walking* and *unsafe–safe traffic,* whereas *vehicle speed* was negatively related to *unsafe–safe traffic* (Figure 8 and Figure 10). The regression equation for the relation between the predictors and the outcome *hinders–stimulates walking* was y = 11.8 − 0.24 *noise* (all *p*-values ≤ 0.006, Adj. R^2^ = 0.11). The regression equation for the other outcome *unsafe–safe traffic* was y = 13.2 − 0.21 *noise* −0.16 *vehicle speed* (all *p*-values ≤ 0.05, Adj. R^2^ = 0.05).

The external validity of these results needs to be commented upon in relation to different subpopulations. In a study where it was examined how different route settings were appraised in terms of plausible or not plausible choices among four groups of cyclists (regular, frequent, occasional, and potential), the order of the results was consistent among the groups. Thus, the principal preferences for different route environment components were the same. However, the sensitivity in the ratings between the groups increased between regular, frequent, occasional, and potential cyclists, when the settings became less attractive. To put it briefly, regular cyclists were less demanding regarding their route environments [30]. If this is translated to the present study, it suggests that regular pedestrians might be less sensitive to high levels of motorized vehicle flow, speed, noise and exhaust fumes. Thus, in the general population, the unstandardised beta coefficients might be higher and the y-intercepts lower.

Another result was that the perception of each traffic variable was negatively related to both the outcome *hinders–stimulates walking*, and to *unsafe–safe traffic.* This means that if imprecise or few traffic variables, e.g., [8,9,10,31] have been included as indicators for traffic, then diverse results can be a consequence. This can be misleading in a search for more critical variables.

How these and other relations can be further analysed will be discussed below. In doing so, we will integrate the results from correlation (CA), multiple regression (MRA), and mediation analyses (MA).

### 4.1. The Relationships between Perceptions of the Predictor Variables of Motor Traffic

There were some high positive correlations between the four traffic variables. The highest was between *vehicle flow* and *noise* (r = 0.77) followed by between *noise* and *exhaust fumes* (r = 0.74).

These high correlations indicate that the analyses of possible separate effects of single traffic variables in relation to *hinders–stimulates walking* and *unsafe–safe traffic* can be hampered when they are jointly involved in MRA analyses. That dilemma can possibly be circumvented with more participants involved, or research strategies that aim to try isolate the effect of separate individual traffic variables. Our approach has been to analyse the relationships between the traffic and outcome variables step by step, with different tools, and thereby hopefully disclosing important information.

### 4.2. Vehicle Speed and Vehicle Flow in Relation to Noise and Exhaust Fumes

The level of vehicle flow is equal to the number of vehicles passing a certain reference point during a given time frame. *Vehicle speed* impacts *vehicle flow*, as was reflected in a correlation of 0.61 and a regression coefficient of 0.72.

*Vehicle speed* and *vehicle flow* were both individually and jointly related to *noise,* whereas only *vehicle flow* was related to *exhaust fumes* in MRA. However, MA showed that *exhaust fumes* was also affected by *vehicle speed* through a mediation via *vehicle flow* (Table 9 and Figure 10). The magnitudes of the mediation from *vehicle speed* via *vehicle flow* to both *noise* (78%) and *exhaust fumes* (95%) were surprisingly high, and have, to the best of our knowledge, not been shown before. Even though perceptions of *exhaust fumes* was unrelated to both of our outcomes, note that in urban areas, pedestrians are frequently exposed to invisible and odourless threats, e.g., particulate matter (PM). 

### 4.3. The Traffic Variables in Relation to the Outcome Variable Hinders–Stimulates Walking

When combining the basic variables *vehicle speed* and *vehicle flow*, and analysing them in MRA, with *hinders–stimulates walking* as an outcome, only *vehicle flow* was negatively related to the outcome. *Noise* had the corresponding role when combining the intermediate outcomes *noise* and *exhaust fumes* in the MRA. Finally, when all traffic variables were included as predictors, only *noise* was negatively related to *hinders–stimulates walking* (Figure 7). The MA showed that both *vehicle flow* and *vehicle speed, per se,* and to about 80%, are mediated via *noise* to *hinders–stimulates walking*. Again, this shows that it is critical which variables are included in the analyses and how the understanding can be developed by using both MRA and MA. Note that we created a composite variable of *vehicle flow* × *vehicle speed* (Table 9). The rationale was to examine if the basic variables combined would have a different effect on the outcome variables compared to when the variables acted on their own. However, the results were in general accordance with those of the single predictors.

Anciaes et al. [9] studied residents close to highly trafficked roads and noted that when both the traffic volume and the traffic speed were rated as high, it appeared that walking was hindered, and the wellbeing was rated lower. This is in line with McGinn et al. (2007) stating that “It is only when high-traffic volume is combined with high-traffic speeds that being physically active outdoors may pose a problem” [32]. However, given the present results those findings could, just as well, be due to higher levels of *noise*, a variable that Anciaes et al. [9] and McGinn et al. [32] did not study. 

As stated, *noise* was related to hindering walking in the present study and has been reported to be negatively associated with walking in other studies, e.g., [11,33,34]. In line with that, transportation noise annoyance can lead to lower levels of physical activity [35]. Proximity to major roads has also been associated with reduced levels of physical activity, independently of noise annoyance [35], and it can possibly be due to a barrier effect of such roads, as has been shown when located between residential and green areas [36].

Both the MRA and the MA analyses point in the direction that *vehicle speed* is an important origin of the deterring effects of *noise* (Figure 10). Speed management should, therefore, be a key issue. In line with that, the World Health Organization suggests a speed limit of 30 km/h in urban areas [37]. Indeed, as is evident from Figure A1 and Figure A2, such an implementation would be important also in the present study area.

The “noise-problem” can only be partially solved with electric cars. There is a thres-hold at approximately 30 km/h [38,39]. Below this speed, an electric car makes less noise than a conventional car (with an internal combustion engine). For speeds above 30 km/h, which are frequent in urban areas, the friction of the tires is the major source of the noise. 

### 4.4. Comments on Relations between Noise and Vehicle flow 

It is noteworthy that the MA indicated that there was a certain portion (19%) of the direct effect from *vehicle flow* on *hinders–stimulates walking,* although not significant. This might be due to its high correlation to noise (r = 0.77), and the not very high level of participants. 

Although *vehicle flow* is highly related to *noise*, we see reasons to point out that our perception of a given vehicle flow can vary to a great extent. The vehicle flow can be more or less visually hidden behind motor vehicles, vegetation, land masses, buildings, etc. (Figure 11a,b). Furthermore, the vehicle flow might be close by or distant, visible in different angles, and moving towards or away from us. Finally, it might be behind us and therefore not visible. Thus, a given vehicle flow, can affect us visually very differently. This is, indeed, in contrast to noise, which reaches us even if we close our eyes. From these perspectives, it is reasonable that *noise* protrudes as a major hindering factor in this study.

The external validity of our results needs to be discussed in relation to both the overall environmental setting and the mode of active transportation. In cross-sectional studies of cyclists in suburban-rural settings, it has been noted that both *vehicle flow* and *noise* were negatively related to *inhibiting–stimulating cycling* [40]. This discrepancy regarding *vehicle flow* in relation to our results can possibly be explained by differences in settings, which may affect some of our perceptions and appraisals fundamentally different. A plausible explanation could be that massive buildings in the inner urban area masks or diminishes the way vehicle flow protrudes compared to in the suburban–rural setting. In support for such an interpretation is an experimental pilot study in a green area, where it was shown that intra-individual comparisons between seeing a high vehicle flow with a high speed on a highway, led to significantly higher (34%) levels of environmental *un*wellbeing compared to when the vehicle flow was masked, despite identical noise levels [41]. We therefore suggest that these relationships should be further studied. One promising way forward is to try to isolate the variables when assessing them. This can be done by stage-managing the assessment site as was done in the pilot study above, where visibility was restricted or non-restricted in situ.

A final comment regarding isolating the effect of vehicle flow on outcomes relates to whether perceptions and appraisals are undertaken with or without daylight. The effect of vehicle lights, in dark surroundings, can be disturbing. Note that, as stated in Methods, the natural light conditions in the present study were good.

### 4.5. The Traffic Variables in Relation to the Outcome Variable Unsafe–Safe Traffic

When combining the basic variables *vehicle speed* and *vehicle flow* in MRA, *vehicle speed* related negatively with *unsafe–safe traffic*, whereas when the intermediate outcomes *noise* and *exhaust fumes* were combined, *noise* related negatively with *unsafe–safe traffic*. Finally, when all traffic variables were included as predictors, both *vehicle speed* and *noise* related negatively to the outcome (Figure 8). Somewhat surprisingly, we noted that the unstandardized beta coefficient for *noise* was 28% higher than for *vehicle speed*. Although this difference was not significant, it deserves attention in future studies.

Interestingly, both *vehicle speed* and *vehicle flow*, respectively, were mediated via *noise* on the outcome *unsafe–safe traffic.* The effect of *vehicle flow* on *unsafe–safe traffic* was fully mediated via *noise* whereas the effect of *vehicle speed* was mediated to a lesser degree.

Safety has been reported as an important issue for pedestrians [11,16,17,42]. Walking has, however, also been reported to be unrelated to route safety and traffic [43]. Still, in the present study, perceptions of both *noise* and *vehicle speed* impacted appraisals of *unsafe–safe traffic* negatively. The observed relation between *noise* and *unsafe–safe traffic* is an unexpected and important finding that demands more attention.

The speed of vehicles directly influences the risk of a crash, as well as the severity of injuries including the likelihood of a fatality [44]. Therefore, the negative relation between *vehicle speed* and *unsafe–safe traffic* is not surprising and is in line with indications that streets with slow traffic are appreciated by those who walk [11].

## 5. Conclusions

*Noise,* which depends on *vehicle flow* and *vehicle speed,* is the dominant negative predictor variable in relation to both the outcome *hinders–stimulates walking* and to *unsafe–safe traffic*. *Vehicle flow* contributed, however, more powerfully to perceptions of *noise* than *vehicle speed* did. *Vehicle speed* had a negative role in relation to *unsafe–safe traffic*. 

The mediation analyses revealed that *vehicle speed* acts on *noise* via *vehicle flow*. *Vehicle speed* protrudes, therefore, as an important origin to the deterring effects of traffic on walking. From the point of promoting walking, speed management and noise reduction should, therefore, be a local, regional, national, and supranational concern.

Finally, this study can further the understanding of how to interpret results from other studies, which may have used fewer traffic variables in an ambition to highlight the influence of the traffic environment. This understanding can assist the research, transport, and planning societies in choosing relevant proxy variables. The results from this study can also be of importance for decision makers and others when aiming to create a sustainable future with higher levels of physical activity within the population, and a decreased local, regional, and global dependency on fossil fuels and electricity. 

To support such aims we have used the results from this study and combined it with other predictors, such as greenery and aesthetics, to further our understanding of which the relevant factors are that deter or facilitate walking [45].

## Figures and Tables

**Figure 1 ijerph-20-03743-f001:**
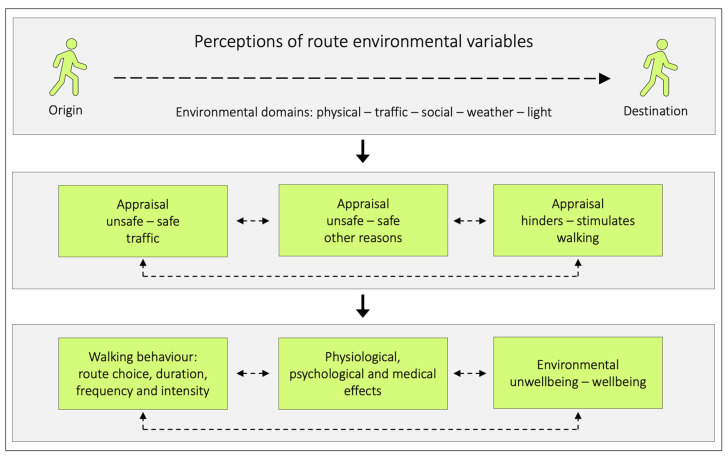
A conceptual model illustrating how perceptions of variables in different route environmental domains may affect pedestrians (modified from [15]). Route environments consist of different environmental domains: the physical environment (stationary objects such as buildings and trees), the traffic environment (mobile objects such as pedestrians, and cars causing e.g., noise, exhaust fumes and particulate matter (PM)), the social environment (interaction between individuals), weather (wind, rain, sun etc.), and light conditions (natural and artificial light). These environmental domains represent a number of predictor variables, whereof some (e.g., noise and exhaust fumes), have the roles of also representing basic variables (e.g., flow and speeds of motor vehicles). They can, thereby, be interpreted as intermediate outcomes. Perceptions of predictor variables in the initial level of the model, can influence how we appraise unsafety–safety due to traffic, and unsafety–safety due to other reasons (such as crime), as well as how we appraise the environment with respect to if it is hindering–stimulating walking. These appraisals can, consecutively, affect our walking behaviour (such as route choice and the amount of walking), have physiological, psychological, and medical effects, and affect our environmentally induced unwellbeing–wellbeing. The bidirectional hatched lines between the boxes indicate potential mutual relationships.

**Figure 2 ijerph-20-03743-f002:**
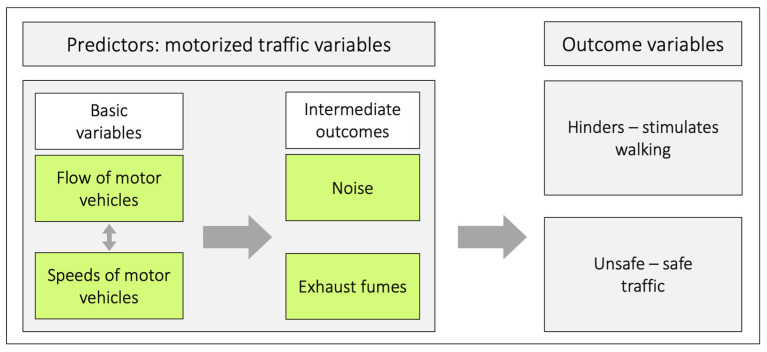
Among the motorized traffic variables, the basic variables generate the intermediate outcomes. All of the traffic variables can, in principle, independently of each other, and in different combinations, relate to the two outcome variables. *Flow of motor vehicles* is equal to the number of vehicles passing a reference point during an established time period. The bidirectional arrow between *flow of motor vehicles* and *speeds of motor vehicles* illustrates that there is a relation between these variables. Under uninterrupted flow conditions, flow, density of vehicles (vehicles × km^−1^) and speed are all related by the following equation: q = k × v where q is *flow*, k is *density*, and v is *speed*.

**Figure 3 ijerph-20-03743-f003:**
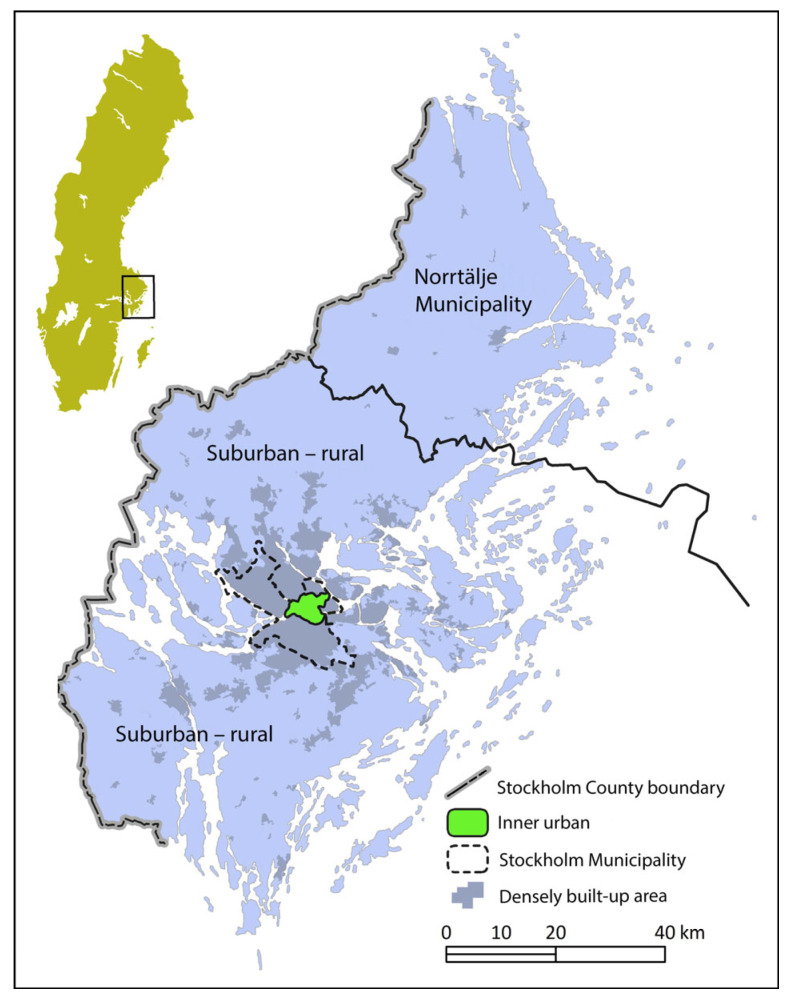
Map over Sweden and Stockholm County, with the inner urban study area of the Municipality of Stockholm in green. The marking for the densely built-up areas illuminates the conditions in 2010. North is at the top of the image.

**Figure 4 ijerph-20-03743-f004:**
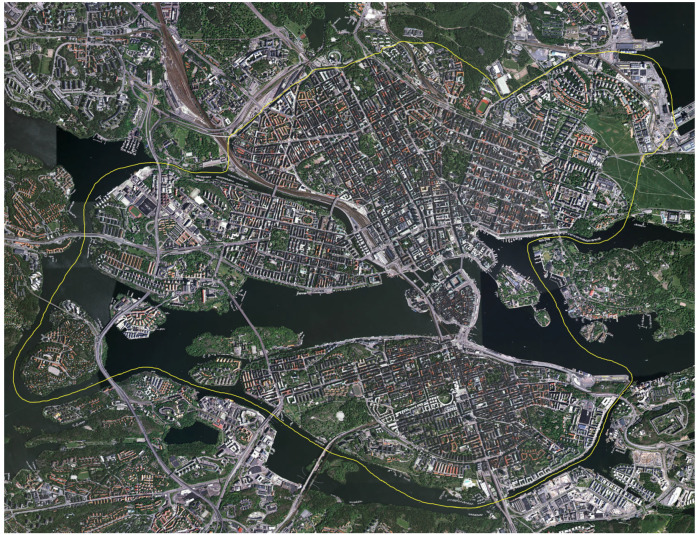
Aerial view of Stockholm in 2005. The area inside the yellow line demarcates the inner urban study area according to our geographical division of the city. North is at the top of the image. (Copyright: Lantmäteriverket, Gävle, Sweden, 2011; Permission 81055230).

**Figure 5 ijerph-20-03743-f005:**
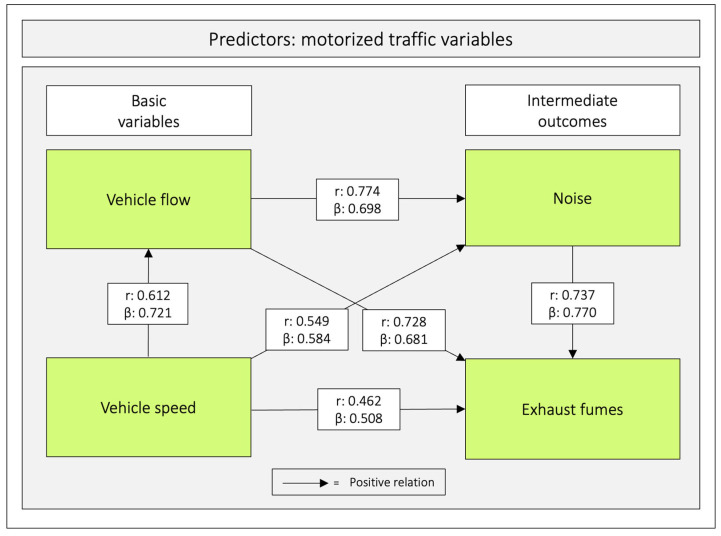
Correlation and regression coefficients between the four predictor variables. Notes: r = Pearson’s correlation coefficient and β = unstandardized beta. The figure is based on Table 4 and Table 5.

**Figure 6 ijerph-20-03743-f006:**
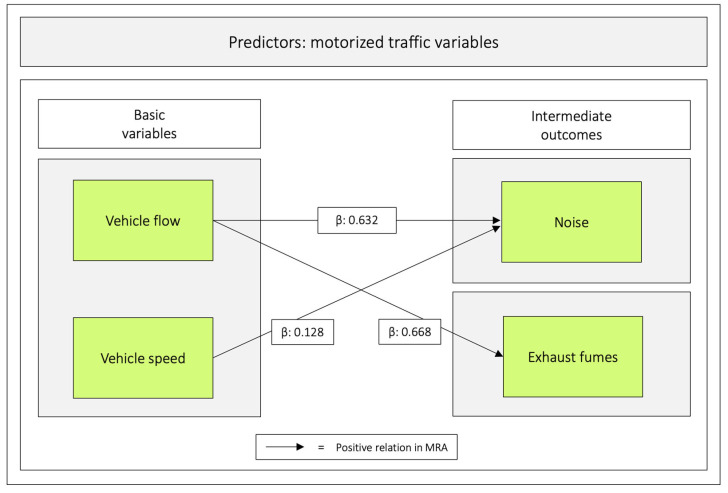
Regression coefficients between *vehicle speed* and *vehicle flow* in relation to *noise* and *exhaust fumes*. Notes: β = unstandardized beta. The figure is based on Table 6.

**Figure 7 ijerph-20-03743-f007:**
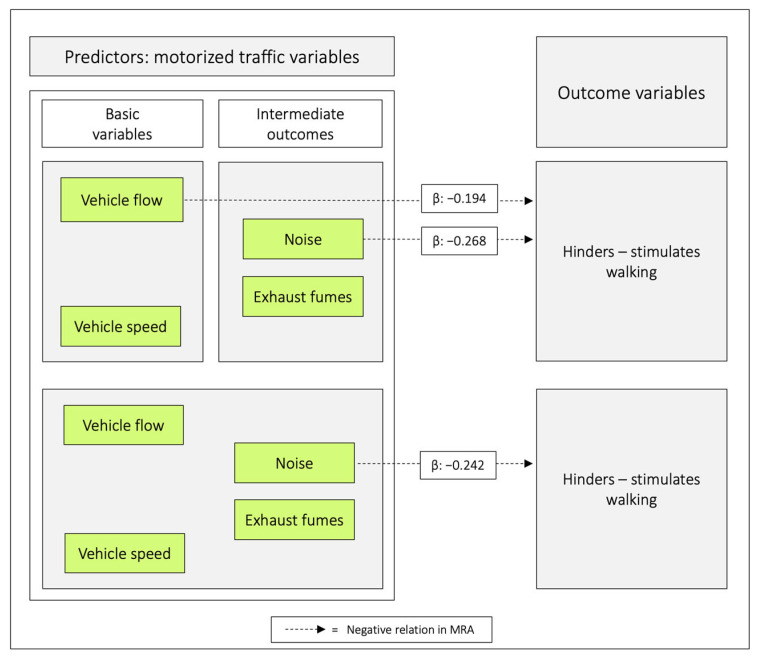
Regression coefficients between combinations of the predictor variables in relation to *hinders–stimulates walking.* Note: β = unstandardized beta. The figure is based on Table 7.

**Figure 8 ijerph-20-03743-f008:**
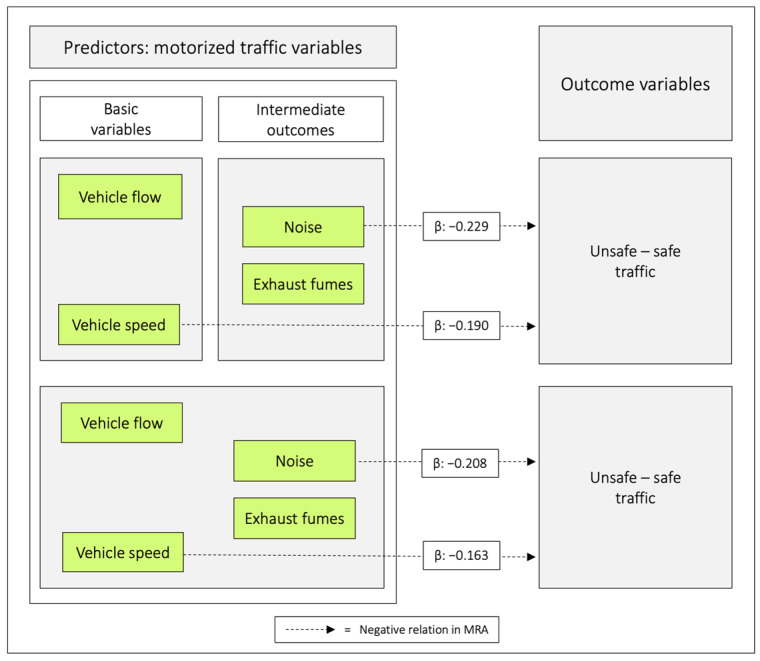
Regression coefficients between combinations of the predictor variables in relation to *unsafe–safe traffic*. Notes: β = unstandardized beta. The figure is based on Table 8.

**Figure 9 ijerph-20-03743-f009:**
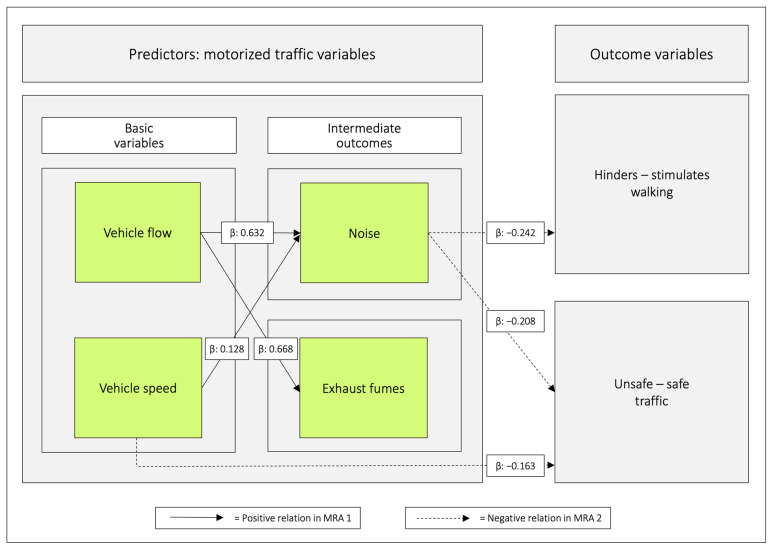
Relations based on multiple regression analysis between the predictor variables (MRA 1), and between the predictor variables and the outcome variables (MRA 2). The solid arrows are based on Table 6. They illustrate that when both *vehicle speed* and *vehicle flow* are analysed jointly as predictors, they relate positively with *noise* whereas only *vehicle flow* relates to *exhaust fumes*. The dashed arrows are based on Table 7 and Table 8. They illustrate that when all traffic variables are analysed jointly as predictors, only *noise* relates negatively to both outcome variables. *Vehicle speed* has the same role in relation to *unsafe–safe traffic*. β = unstandardized beta.

**Figure 10 ijerph-20-03743-f010:**
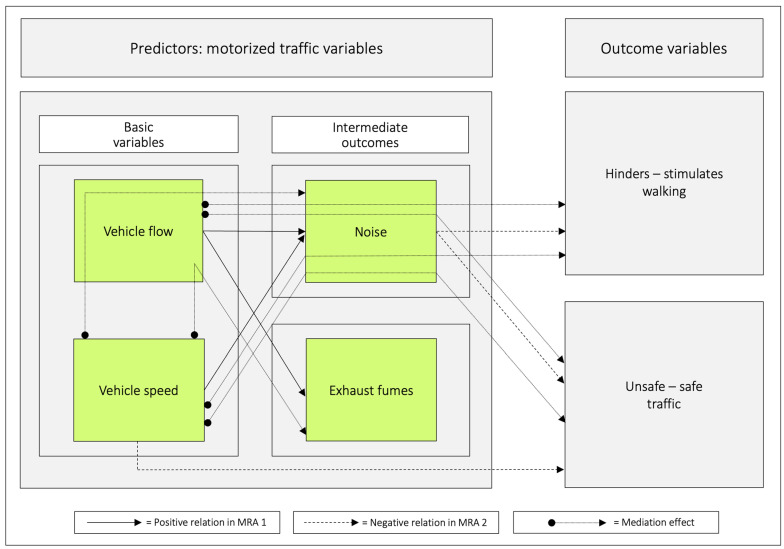
The solid arrows between the predictor variables are based on the multiple regression analyses in Table 6 (MRA 1). They illustrate that when both *vehicle speed* and *vehicle flow* were analysed jointly as predictors, they related positively with *noise* whereas only *vehicle flow* related to *exhaust fumes.* The dotted arrows are based on the mediation analyses in Table 9. They illustrate that *vehicle speed* had an indirect effect on both *noise* and *exhaust fumes*, and that both *vehicle speed* and *vehicle flow* have indirect effects on both outcome variables via *noise*. The dashed arrows are based on Table 7 and Table 8 (MRA 2). They illustrate that when all traffic variables are analysed jointly as predictors, only *noise* relates negatively to both outcome variables. *Vehicle speed* had the same role in relation to *unsafe–safe traffic.* Given these different roles for *vehicle speed*, it protrudes as an important origin to the deterring effects of traffic on walking.

**Figure 11 ijerph-20-03743-f011:**
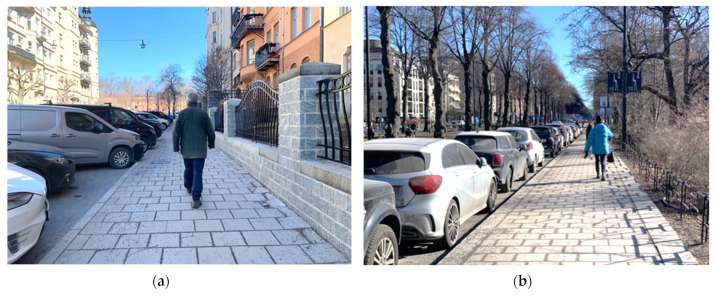
(**a**,**b**) Two photos from the inner urban study area of Stockholm in March 2022. The predominant street network is laid out in a grid pattern with a mixture of different streets. Cars are often parked along streets protecting the pedestrians (physically) by providing a barrier that sometimes will hide the vehicle flow and distance the pedestrians from the vehicles. However, noise will reach those on the pavement regardless of the barrier provided by the vehicles. When data were collected in May 2005, the trees had leaves. Photo: Dan Andersson.

**Table 1 ijerph-20-03743-t001:** Descriptive characteristics of participants (*n* = 282–294) *.

Background Factors	
Females **, %	77
Age in years **, mean ± SD	49.5 ± 10.4
Weight in kg, mean ± SD	68.4 ± 10.6
Height in cm, mean ± SD	171.1 ± 8.2
Body mass index, mean ± SD	23.3 ± 2.7
Gainful employment, %	97
Educated at university level **, %	79
Income **:	
≤25,000 SEK *** a month, %	37
25,001–30,000 SEK *** a month, %	27
≥30,001 SEK *** a month, %	35
Participant and both parents born in Sweden, %	80
Having a driver’s licence, %	89
Usually access to a car, %	56
Leaving home 7–9 a.m. to walk to place of work or study, %	80
Leaving place of work or study 4–6 p.m. to walk home, %	73
Number of walking commuting trips per year ****, mean ± SD	278 ± 162
Overall physical health either good or very good, %	77
Overall mental health either good or very good, %	85

Notes: * 294 individuals have complete data regarding ACRES and the four background variables that are used in the analyses; ** Included in the multiple regression analyses and in the mediation analyses; *** SEK = Swedish crowns (SEK) 2005: €1 ≈ 9 SEK; US$1 ≈ 8 SEK; **** The number of walking commuting trips is based on 224 participants, whereof four had remarkably high values, which therefore were substituted with mean values for the remainder of the participants. The low response rate is due to missing values in at least one of the 12 months leading to exclusion in the sum score.

**Table 2 ijerph-20-03743-t002:** The applied items from the ACRES assessing pedestrian’s perceptions of the environment.

		Question	15-Point Response Scale		Variable Name
			1	15	
**Environmental Variables**	**Outcome Variables**	Do you think that, on the whole, the environment you walk in hinders/stimulates your commuting?	Hinders a lot	Stimulates a lot	Hinders–stimulates walking
		How unsafe/safe do you feel in traffic as a pedestrian along your route?	Very unsafe	Very safe	Unsafe–safe traffic
	**Predictor** **Variables**	How do you find the speeds of motor vehicles (taxis, lorries, ordinary cars, buses) along your route?	Very low	Very high	Speeds of motor vehicles
		How do you find the flow of motor vehicles (number of cars) along your route?	Very low	Very high	Flow of motor vehicles
		How do you find the noise levels along your route?	Very low	Very high	Noise
		How do you find the exhaust fume levels along your route?	Very low	Very high	Exhaust fumes

Notes: This is a translation of the original ACRES in Swedish. Regarding the two variables *speeds of motor vehicles* and *flow of motor vehicles*; from now and onwards we will use the truncated terms: *vehicle speed* and *vehicle flow*.

**Table 3 ijerph-20-03743-t003:** Perceptions of the environmental variables in males and females (mean, SD, and (95% CI)). For explanation of ratings of the scale, see Table 2.

	Outcome Variables		Predictor Variables			
	Hinders–Stimulates Walking	Unsafe–Safe Traffic	Vehicle Speed	Vehicle Flow	Noise	Exhaust Fumes
Men (*n* = 69)	10.4	11.0	8.90	9.80	9.51	9.30
	3.04	3.18	2.74	3.30	3.34	3.45
	(9.70–11.2)	(10.2–11.7)	(8.24–9.56)	(9.01–10.6)	(8.71–10.3)	(8.47–10.1)
Women (*n* = 225)	10.4	10.8	9.77 *	10.3	9.98	9.88
	2.95	3.47	3.15	3.76	3.26	3.45
	(10.1–10.8)	(10.4–11.3)	(9.36–10.2)	(9.83–10.8)	(9.55–10.4)	(9.43–10.3)

Note: * Significant difference (*p* < 0.05).

**Table 5 ijerph-20-03743-t005:** Relations between the predictor variables.

Outcome	y-Intercept (95% CI)	*p*-Value	Predictor	Regression Coefficient, Unstandardized B (95% CI)	*p*-Value	Adj. R^2^
Noise	4.32	<0.000	Vehicle speed	0.584	<0.000	0.292
	(2.26–6.38)			(0.480–0.688)		
Noise	2.00	0.011	Vehicle flow	0.698	<0.000	0.595
	(0.46–3.53)			(0.631–0.764)		
Exhaust fumes	6.69	<0.000	Vehicle speed	0.508	<0.000	0.213
	(4.40–8.97)			(0.392–0.623)		
Exhaust fumes	3.72	<0.000	Vehicle flow	0.681	<0.000	0.526
	(1.97–5.47)			(0.606–0.757)		
Exhaust fumes	4.00	<0.000	Noise	0.770	<0.000	0.547
	(2.32–5.69)			(0.688–0.852)		
Vehicle flow	4.61	<0.000	Vehicle speed	0.721	<0.000	0.373
	(2.46–6.77)			(0.612–0.830)		

Notes: The background variables *sex*, *age*, *education*, and *income* were included in the analysis. Only significant (*p* < 0.05) background variables and their level of significance are reported. The background variable *age* was significant in two models; *exhaust fumes* (as an outcome) and *vehicle speed* (as a predictor): unstandardized B = −0.035 (*p* = 0.047) and *exhaust fumes* (as an outcome) and *noise* (as a predictor): unstandardized B = −0.031 (*p* = 0.019).

**Table 9 ijerph-20-03743-t009:** Mediation analyses between the four predictor variables of motor traffic (MA1–MA4), between the same predictor variables and the two outcome variables (MA5–MA8), as well as between the composite variable (*vehicle flow* × *vehicle speed*) and the outcome variables (MA9–MA10).

Model	Predictor (X)	Mediator (M)	Outcome (Y)	Standardized Total Effect of X on Y		Standardized Direct Effect of X on Y		Standardized Indirect Effect of X on Y		
				b	*p*-Value	b	*p*-Value	b	95% CI	% of Total Effect
MA1	Vehicle flow	Vehicle speed	Noise	0.779	<0.000	0.706	<0.000	0.073	0.010–0.140	9
MA2	Vehicle speed	Vehicle flow	Noise	0.549	<0.000	0.120	0.011	0.428	0.343–0.513	78
MA3	Vehicle flow	Vehicle speed	Exhaust fumes	0.721	<0.000	0.707	<0.000	0.014	−0.049–0.077	2
MA4	Vehicle speed	Vehicle flow	Exhaust fumes	0.452	<0.000	0.023	0.657	0.429	0.343–0.515	95
MA5	Vehicle flow	Noise	Hinders–stimulates walking	−0.276	<0.000	−0.054	0.543	−0.223	−0.366–−0.090	81
MA6	Vehicle speed	Noise	Hinders–stimulates walking	−0.206	<0.000	−0.037	0.574	−0.168	−0.248–−0.100	82
MA7	Vehicle flow	Noise	Unsafe–safe traffic	−0.183	0.002	0.007	0.939	−0.190	−0.324–−0.042	104
MA8	Vehicle speed	Noise	Unsafe–safe traffic	−0.220	<0.000	−0.127	0.066	−0.093	−0.172–−0.019	42
MA9	Composite variable	Noise	Hinders–stimulates walking	−0.266	<0.000	−0.065	0.411	−0.201	−0.319–−0.099	76
MA10	Composite variable	Noise	Unsafe–safe traffic	−0.229	<0.000	−0.118	0.148	−0.111	−0.229–0.008	48

Notes: Covariables included in the analyses were *sex*, *age*, *education*, and *income*. *Age* was significant in model MA4 with respect to the total effect of X on Y.

## Data Availability

Data is available upon request.

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
