# Peer review of "Pedestrians’ Perceptions of Motorized Traffic Variables in Relation to Appraisals of Urban Route Environments"

_ijerph, 2023, doi:10.3390/ijerph20043743_

Round 1
Reviewer 1 Report
My comments are given as follows:
(1) The mean, SD and 95% CI should be given before each row in Table 3.
(2) For the readability, the tables and figures in the Appendix should be given in the main body of the manuscript, because they are the important supports for the main findings.
(3) In section 3, the brief explanation should be given to each sub-section about the analysis result (i.e., the data listed in the tables).
Reviewer 2 Report
The paper covers an interesting topic. My concerns about the manuscript are as follows:
*) The data used in this study dates back to 2004/2005. Actually, as far as I understand, in the present paper, same dataset that has been used to develop the ACRES tool has been used. Obviously, in 20 years, things changed a lot, especially in urban areas. It is not only the urban environment but also the individuals (their perceptions, way of living, etc.) that have probably changed. Of course, it would be too much to ask the authors to compile new data. However, I believe, it is necessary to discuss the probable impact of the changes in the last 20 years in the paper (with a paragraph or two).
*) Connected with the point above, there was (and for some, still is) the impact of the pandemic going on. Interestingly, the paper totally disregards the pandemic; there is not a single mention about it. I understand that the data is from earlier time and the dynamics of walking or using active modes in general might not change a lot because of the pandemic. However, this must be discussed in the paper (again, with a paragraph or two).
*) I believe it is a question mark whether figures and/or maps presented in the paper are aligned with the dataset. For instance, Figure 4 that contains the aerial view of Stockholm is 2011 dated; 7 years after the data was collected. Moreover, Figure 3 and 5 are not dated. Again, I am not asking for 2004 figures or maps but the discrepancy in years is needed to be mentioned.
*) The authors refer to their earlier studies a lot throughout the manuscript. I am not saying that they are all inappropriate. However, this makes me wonder whether this submission should be assumed as a follow-up study or a standalone one. To be more specific, let’s consider the first paragraph of “Section 2.4 Study Area”. There, the authors give very brief information about the study area (mainly population at the time of the study and the geographical divisions of Stockholm) and state that “A more detailed description of the study area can be found in Wahlgren and Schantz (2011, 2012)”. I reviewed those papers (both published in BMC Medical Research Methodology journal) and saw that there are much more details in them, as the authors mentioned. Furthermore, in the 2011 dated paper, the study area is explained in two separate sections for inner urban and suburban areas, respectively. In addition, in both, there is an aerial view from 2005 (I think using this in the present paper is better, as mentioned before). Now the question is what is the limit to refer earlier studies? Personally, I believe if it is the methodology that have been used or some index that have been developed before, this could be normal. For example, the authors talk about the ACRES tool and include Figure 1 and Section 2.3.1 to summarize the tool. This is fine because, the topic of this submission is not about how ACRES is obtained. If I want to learn about this in details, I can review the cited articles. However, on the other hand, I believe study area can not be treated as the same. It is eventually about where the study is conducted, good knowledge about the study area would improve the understanding and therefore, this information should be given in the present paper. For me, this is similar to not including Table 1 (Descriptive characteristics of participants) in the paper (and referring to an earlier study). I do not think the author would have done that (and they did not). As a result, it is much better to give more details about the study area.
*) I think basic information about the values of the variables given in Table 5-7 is needed. For example, according to the exhaust fumes model in Table 5, y = 3.60 + 0.668 * Traffic Flow + 0.026 * Vehicle Speed. Maybe I skipped something but, it is not clear what is taken as “traffic flow” and “vehicle speed” and what is “y”. Probably, the authors would say that Table 2 explains all. From the table (and the explanation above the table), it is understood that there are scores between 1 and 15 and these scores are inserted to the equation. In turn, y will be a score as well. Then, if I am not wrong, all the regression analyses will give a satisfaction (or a perception) score. From this, I can say that naming them as flow or speed is a bit problematic. As a rule of thumb, in academic journals, the tables should be self-explanatory. In the current presentation of the variables, they seem not to be because when one variable is named as the traffic flow, readers might think it is the actual flow (veh/hr). However, as far as understand, it is not and it is the perception score. As a result, I recommend the authors to name the variables as Traffic Flow Score, Exhaust Fumes Scores, etc. to avoid confusions.
*) I also would like to object the presentation in Figures A4 and A5 in the appendix. For instance, in Figure A4, only the significant coefficients in Table 6 are given. There are three significant coefficients in the table. For the model that contains all the predictor variables only traffic noise is significant and the coefficient equals to -0.242. The depiction in Table 6 reports this and does not report anything else about the remaining three coefficients (because they are insignificant). Assume a case when the insignificant predictor variables are removed from the model and only traffic noise is kept. Would the coefficient be equal to -0.242? The answer is no and I believe the current presentation of Figure 6, gives the impression that -0.242 is from a model that contains traffic noise only. Normally, when there are insignificant coefficients in a model, one can not disregard them and directly use the coefficient estimated for the significant model. If there is removal then a new model is needed. Based on my explanation above, I strongly recommend the authors to remove Figure A4, A5 and other figures that might cause a similar confusion. Please do not get me wrong, I am not against a model that includes insignificant coefficient estimates. I am objecting the presentation format in the figures.
Author Response
Please ses the enclosed response letter.
